# Analysis of the Uniformity of Mechanical Properties along the Length of Wire Rod Designed for Further Cold Plastic Working Processes for Selected Parameters of Thermoplastic Processing

**DOI:** 10.3390/ma17040905

**Published:** 2024-02-15

**Authors:** Konrad Błażej Laber

**Affiliations:** Department of Metallurgy and Metal Technology, Faculty of Production Engineering and Materials Technology, Czestochowa University of Technology, 19 Armii Krajowej Ave., 42-200 Czestochowa, Poland; konrad.laber@pcz.pl; Tel.: +48-34-325-07-97

**Keywords:** wire rod rolling, metallographic analysis, thermovision investigation, mechanical properties, steel for cold upsetting

## Abstract

This study presents the results of research, the aim of which was to analyze the uniformity of the distribution of selected mechanical properties along the length of a 5.5 mm diameter wire rod of 20MnB4 steel for specific thermoplastic processing parameters. The scope of the study included, inter alia, metallographic analyses, microhardness tests, thermovision investigations, and tests of the wire rod mechanical properties (yield strength, ultimate tensile strength, elongation, relative reduction in area at fracture), along with their statistical analysis, for three technological variants of the rolling process differing by rolling temperature in the final stage of the rolling process (Reducing Sizing Mill rolling block [RSM]) and by cooling rate using STELMOR^®^ cooling process. The obtained results led to the conclusion that the analyzed rolling process is characterized by a certain disparity of the analyzed mechanical properties along the length of the wire rod, which, however, retains a certain stability. This disparateness is caused by a number of factors. One of them, which ultimately determines the properties of the finished wire rod, is the process of controlled cooling in the STELMOR^®^ line. Despite technological advances concerning technical solutions (among them, increasing the roller track speed in particular sections), it is currently not possible to completely eliminate the temperature difference along the length of the wire rod caused by the contact of individual coils with each other. From this point of view, for the analyzed thermoplastic processing parameters, there is no significant impact by the production process parameters on the quality of the finished steel product. Whereas, while comparing the mechanical properties and microstructure of the wire rod produced in the different technological combinations, it was found that the wire rod rolled in an RSM block at 850 °C and cooled after the rolling process on a roller conveyor at 10 °C/s had the best set of mechanical properties and the smallest microstructure variations. The wire rod produced in this way had the required level of plasticity reserve, which enables further deformation of the given type of steel in compression tests with a relative plastic strain of 75%. The uniformity of mechanical properties along the length of wire rods designed for further cold plastic working processes is an important problem. This is an important issue, given that wire rods made from 20MnB4 steel are an input material for further cold plastic working processes, e.g., for the drawing processes or the production of nails.

## 1. Introduction

The quality of a wire rod is mainly determined by the temperature–strain rolling parameters in the finishing rolling blocks and the cooling rate of the wire rod on the roller conveyor using blown air. Rolling in finishing blocks is a complex process due to the interaction between metal and tools in the rolling mill stands having a common drive and the occurrence of inter-stand forces. High rolling speeds and small distances between stands in the rolling blocks determine the material’s strengthening and softening processes. High rolling speeds also affect the heat transfer between the roll, the tools, and the surroundings, as well as the end-of-rolling temperature, which, together with subsequent cooling, influences the microstructure and properties of the finished product [1].

The uniformity of the mechanical properties along the length of the rolled band depends on a number of factors. Among the most important is the temperature distribution along the length of the rolled material throughout the whole rolling line, from the heating furnace to the cooling process of the finished product. The influence of the uniformity of the temperature distribution along the length of the feedstock on the properties of the finished product was discussed by the authors of the study, among others [2]. In their study, they proposed, among other things, ways to reduce differences in temperature by using special heat covers. According to the results presented in [2], it was determined, inter alia, that the temperature uniformity along the processed wire rod has a significant effect on the mechanical properties of the finished product. Uniformity of temperature along the length of the rolled band also influences the plasticity of the metal’s flow and, thus, the dimensional accuracy of the finished product, as evidenced by the results of studies presented, among others, in [3]. Divergences in the temperature distribution along the length of the rolled metal also affect the energy and force parameters of the rolling process. With the view of reducing temperature inhomogeneity along the length of the rolled band, the authors of [3] proposed a solution consisting of heating in the furnace to a higher temperature during the final stretch of the feedstock.

An important stage determining the uniformity of the mechanical properties of the wire rod along its length is the process of controlled cooling in the STELMOR^®^ line. This issue was researched by the authors of, inter alia, [4,5]. The results obtained indicated that, among other factors, the nonuniformity of the mechanical properties of the wire rod along its length is due to the characteristic arrangement of the coils (in the form of a “spiral”) during the cooling process. As a result of the contact of the individual coils with each other, areas of varying (elevated) temperature, so-called ‘hot spots’, are created, which result in different microstructure development during cooling and consequently in heterogeneity of the mechanical properties of the finished product. The authors of [4,5] indicated ways to reduce the distinction in mechanical properties along the length of the wire rod by, among other things, changing the density of the wire coils’ setting on the roller conveyor. In addition, [4] proposed a comprehensive model for predicting the properties of wire rods.

The authors of [6,7,8,9,10,11,12] also dealt with issues related to heat transfer during controlled cooling of wire rods on a roller conveyor. These works can be used in detailed studies of the conditions for cooling the wire rod, e.g., to improve the uniformity of the wire rod properties along its length. Study [8] presents a model for simulating the cooling process of wire rods on a roller conveyor. The study takes into account all types of heat transfer as well as coils’ density. A system for testing and recording the temperature of wire rods during cooling on a roller conveyor is presented in [6]. It was used during industrial research aimed, inter alia, at determining the interdependence between the cooling rate and the mechanical properties of the finished product. Study [12] presents a system for controlling the wire rod attributes during cooling on a roller conveyor based on time–temperature-transformation (TTT) diagrams.

Despite technological advances in the applied technical solutions (such as increasing the roller track speed in particular sections, controlling the coil density of the wire rod, air distribution improving cooling uniformity, new nozzle sets bettering cooling efficiency, or adding fans of higher capability and equalizing chambers) presented in, inter alia, [13], complete elimination of the temperature difference along the length of the wire rod caused by the contact of individual coils with each other constitutes an unresolved problem.

In the technical literature on the rolling process, it is possible to find studies dealing with the influence of the production process parameters on the quality (mainly microstructure and mechanical properties) of the finished product [14,15,16,17,18,19,20,21,22,23,24,25]. Several groups of steels for wire rod production were described in [14], such as interstitial free (IF) ferritic steels, ferritic/martensitic steels, and pearlitic micro-alloyed steels. The possibilities of forming the microstructure of these steels are described, as well as the basic technological guidelines for obtaining a finished product with the desired attributes. Study [20] is mainly concerned with the possibility of improving the properties of low-carbon steel wire rods by introducing alloy additions into the steel, which enhance the mechanical properties of the finished product. The authors of this study have shown that an important factor in terms of the range of applications of the obtained wire rod is the significant increase in its capacity for further direct cold plastic working processes. Studies [15,16,17,18,19,21] deal with the rolling processes of wire rods made from high-carbon steels. In these studies, the effects of temperature and cooling conditions on the microstructure and properties of the wire rod were analyzed. Studies [26,27] present a model of the microstructure development during the rolling of wire rods made of high-carbon steel, type C70D, and the test results of using it.

However, few studies analyze the quality of the finished rolled product in terms of uniformity of microstructure and mechanical properties along its length, especially with regard to wire rods. A brief analysis of the quality of the finished product concerning microstructure and mechanical properties along its length can be found in [28], among others. However, this study deals with smooth round bars.

For this reason, the undertaken research topic, that is, the analysis of the uniformity of mechanical properties along the length of cold upsetting steel wire rods for selected thermoplastic processing parameters, is, in the author’s opinion, valid. This is an important issue, given that wire rods made from 20MnB4 steel are an input material for further cold plastic processing, e.g., for the drawing industry or the production process of nails.

## 2. Materials and Methods

### 2.1. Materials

The tests presented in this study were carried out on a 5.5 mm diameter wire rod made of low-carbon, cold-upsetting steel of the 20MnB4 type with a chemical composition in accordance with the PN-EN 10263-4:2004 standard (Table 1) [29].

### 2.2. Characteristics of the Wire Rod Rolling Process—Main Process Parameters of Thermoplastic Processing

This study was carried out for an exemplary rolling mill of a combined type (combination of bar mill and wire rod mill). The input material for the rolling process was an ingot from the continuous casting process with a square cross-section, side of 160 mm, and length of 14.000 mm. The rolling process for the wire rod of 20MnB4 steel, with a final diameter of 5.5 mm, in the continuous rolling mill took place in 17 rolling passes, while the rolling in the wire rod mill took place in two rolling blocks: a 10-rolling stand NO-TWIST MILL (NTM) and 4-rolling stand REDUCING SIZING MILL (RSM).

The analyses presented in this study were carried out for three technological variants (V1–V3), differing both in the rolling temperature in the RSM rolling block for the wire rod and in the cooling rate of the finished product on the roller conveyor in STELMOR^®^ technology. Table 2 and Table 3 present the most important thermoplastic processing parameters for the whole process, studied, determined, and verified in studies [30,31,32], among others.

**In variant one (V1)**, the temperature of the wire rod at the entrance to the RSM rolling block (rolling pass no. 28) was approximately 800 °C. After the processed material exited the RSM rolling block, it was cooled with water under high-pressure to 750 °C for 0.3 s at a rate of 166.67 °C/s, using an accelerated cooling system. Further cooling in air took place on a roller conveyor at a rate of 0.4 °C/s, to a temperature of about 575 °C and then at a rate of 1.3 °C/s to a temperature of ca. 200 °C (heat insulating covers in the STELOR^®^ line closed, fans switched off).

**In variant two (V2)**, the temperature of the wire rod at the entrance to the RSM rolling block (rolling pass no. 28) was approximately 850 °C. After the processed material exited the RSM rolling block, it was cooled with blown air at a rate of 5.0 °C/s to a temperature of approximately 500 °C and then at a rate of 1.0 °C/s to a temperature of approximately 200 °C (thermal insulation covers in the STELOMR^®^ line open, fans switched off).

**In variant three (V3)**, the temperature of the wire rod at the entrance to the RSM rolling block (rolling pass no. 28) was approximately 850 °C. After the processed material exited the RSM rolling block, it was cooled with blown air at a rate of 10.0 °C/s to a temperature of approximately 500 °C and then at a rate of 1.0 °C/s to a temperature of approximately 200 °C (thermal insulation covers in the STELOMR^®^ line open, fans switched on—rotational speed 75% of maximum).

### 2.3. Methods

So as to obtain a finished product with a uniform fine-grained ferritic–pearlitic microstructure without clear-marked banding, the final deformation stage (rolling in the RSM rolling block) should take place in the single-phase (austenitic) range, when its temperature is 30 ÷ 80 °C higher than the temperature at the beginning of the austenite transformation Ar_3_ [1,33,34,35,36]. For this purpose, numerical modeling of the process under analysis was carried out using QTSteel^®^ software (ITA Ltd., Ostrava, Czech Republic, Metaltech Services Ltd., Gateshead, UK, release 3.4.1) based on the finite element method [37]. This stage of the study aimed to determine the temperatures of phase transformations and the contribution of the individual phases of the microstructure to the thermoplastic processing parameters under analysis.

In addition (for verification), physical modeling of the analyzed wire rod rolling process using non-free torsion was carried out by using STD 812 torsion plastometer (manufactured by Bähr Thermoanalyse GmbH Hüllhorst, Germany, now TA Instruments New Castle, DE, USA), according to the methodology presented in [32]. The purpose of this stage of research was to determine (check) the microstructure of the steel under study immediately before the deformation process in the RSM rolling block. Thus, the tested material was hardened right before the deformation stage, simulating rolling the wire rod in the RSM rolling block. Then, the metallographic analysis and microhardness measurements of the resulting microstructure were carried out.

The subsequent stage of the research focussed on checking the uniformity of the temperature of the rolled band along its length in the rolling line. These tests were carried out using the ThermaCAM SC640 (FLIR Systems, Wilsonville, OR, USA) thermovision camera, which is equipped with an uncooled detector [38]. ThermaCAM Researcher Professional 2.10 (FLIR Systems AB) software was used to process these data.

This was followed by testing selected mechanical properties of the finished wire rod along its length (including their statistical analysis, by using Statistica ver. 13, TIBCO Software Inc., Palo Alto, CA, USA), produced according to technological variants V1–V3. These tests were carried out in a static tension test in accordance with the standard [39] using a Zwick Z/100 materials testing machine (produced by ZwickRoell, Wroclaw, Poland) [40]. In order to assess the capacity of the tested steel for further cold processing, upsetting tests were carried out according to the standard [41].

The scope of this study also included microstructure analysis (measurement of ferrite grain size and hardness using Vickers’ method—loading force 9.81 N, loading time 5 s) of the wire rod along its length (in longitudinal section).

All metallographic analyses were conducted with light microscopy using a Nikon Eclipse MA 200 microscope (Nikon Metrology NV, Leuven, Belgium) with NIS-Elements software [40]. Hardness measurements were conducted using the Vickers’ method with a FutureTech FM 700 microhardness tester (Kawasaki, Japan) [40].

### 2.4. Numerical Modelling—Mathematical Model of QTSteel^®^ Software

In the QTSteel^®^ program, when modeling the microstructure and mechanical properties of heat-treated or thermomechanical-treated steel, data from the cooling curves on the TTT diagram are used. Calculating the percentage content of the microstructure components is performed successively for the relevant sections of the cooling curve. To describe the kinetics of the transformation of individual components of the microstructure, the program uses the Avrami Equation (1) [42,43]:(1)$$X_{i}(T,t)=(1-\exp(-k(T)\times t^{n(T)}))\times X_{\gamma },$$
where: $X_{i}(T,t)$—volume fraction of individual components of the microstructure: ferrite, perlite, bainite, *k*(*T*) and *n*(*T*)—parameters depending on the transformation mechanism and places of privileged nucleation and on the cooling rate, calculated based on TTT charts for a given temperature, *T*—temperature, *t*—time,—volume fraction of residual austenite.

The volume fraction of martensite during martensitic transformation is calculated by using the Koistinen–Marburger equation [37,43]:(2)$$X_{m}(T)=(1-\exp(-b\times {\left(T_{ms}-T\right)}^{n}))\times X_{\gamma },$$
where: $X_{m}$—volume fraction of martensite, *b*, *n*—constant, $T_{ms}$—martensitic transformation start temperature, *T*—temperature, $X_{\gamma }$—volume fraction of residual austenite.

Vickers *HV* hardness is determined by means of a regression equation [42,43]:(3)$$HV=C_{0}+X_{f}\times \sum (D_{i}\times c_{i}+X_{p}\times \sum E_{i}\times c_{i}+X_{b}\times \sum F_{i}\times c_{i}+X_{m}\times \sum G_{i}\times c_{i},$$
where: *HV*—Vickers hardness, $X_{f}$, $X_{p}$, $X_{b}$, $X_{m}$—volume fractions: ferrite, perlite, bainite, martensite, *C*_0_, *D_i_*, *E_i_*, *F_i_*, *G_i_*—constant, *c_i_*—the percentage of alloying additions.

The tensile strength was determined based on Equation (4) [42]:(4)UTS=f(HV)=−a+b×HV,
where: *UTS*—Ultimate tensile strength, *HV*—Vickers hardness, and *a*, *b*—constant.

Yield strength *YS* is determined by Equation (5) [42,43]:(5)$$YS=f(D_{\alpha },C_{r},X_{f},\sum \left(X_{p}+X_{b}+X_{m})\right),$$
where: *D_α_*—ferrite grain size, *C_r_*—cooling rate, $X_{f}$, $X_{p}$, $X_{b}$, $X_{m}$—volume fractions: ferrite, perlite, bainite, martensite.

Accurate results of research carried out using the DIL 805 A/D dilatometer, the aim of which was to develop TTT and DTTT diagrams and to determine the best cooling conditions for 20MnB4 steel, were published in [44]. According to published research, in the case of cooling wire rods on a roller conveyor, greater accuracy in predicting the microstructure and mechanical properties of the finished product is achieved based on DTTT charts, which take into account the deformation process preceding the cooling of the rolled band.

Taking into account the obtained results, the DTTT graph was used to determine the influence of the cooling conditions on the forming of the wire rod microstructure immediately after the deformation process. It was concluded that in order to obtain a ferritic–pearlitic microstructure in the finished product, the cooling rate should not exceed 15 °C/s. Faster cooling causes the formation of bainite, bainitic–martensitic, and martensitic structures in the material, which results in a decrease in the ability of the investigated steel for further cold plastic working processes or, in extreme cases, prevents it.

Numerical modeling using QTSteel software was carried out using input data shown in Table 2 and Table 3 and published in [30].

## 3. Results and Discussion

### 3.1. Numerical Modelling Results—QTSteel^®^ Software

Figure 1 shows part of the DTTT (Deformation Time Temperature Transformation) graph with phase transformation curves and cooling curves according to variants V1–V3, obtained as a result of numerical modeling of the thermoplastic processing of the analyzed rolling process for 5.5 mm diameter wire rods of 20MnB4 steel.

The percentage amounts of microstructure components, hardness, selected mechanical properties, and characteristic temperature values—obtained from numerical modeling of the thermoplastic processing of a 5.5 mm diameter wire rod made from 20MnB4 steel—are shown in Table 4.

According to the analysis of the test results obtained from numerical modeling of thermoplastic processing of a 5.5 mm diameter wire rod made from 20MnB4 steel according to variant V1, it was ascertained that the transformation of austenite to ferrite during cooling (Ar_3_) began at the temperature of 744 °C. In contrast, the temperature of the onset of the transformation of austenite to pearlite during cooling (Ar_1_) was 655 °C. The temperatures at the beginning and completion of ferrite to austenite transformation during heating were correspondingly: 719 °C and 827 °C. Bearing in mind the earlier stages of rolling (multi-sequence deformation over a wide temperature range, multi-stage accelerated cooling and reheating due to heat conduction from the center of the material towards its surface and as a result of deformation at high rates) [30] as well as the accelerated cooling after rolling in the RSM block, and upon analyzing the cooling curve (Figure 1) while taking into account the phase transition temperatures, it can be concluded that the rolling and cooling of steel in this variant took place in the single-phase range but relatively close to the temperature of the beginning of austenite to ferrite (Ar_3_) transformation.

Under these thermoplastic processing conditions, percentage amounts of ferrite and pearlite were 94.5% and 5.5%, respectively. The average hardness value was 173 *HV*. The mechanical properties of the material formed under these conditions were yield strength—319 MPa and ultimate tensile strength—516 MPa. The plasticity reserve amounted to 0.618.

Analysis of the results of numerical modeling according to the variant V2 led to the conclusion that the temperature at the onset of the transformation of austenite to ferrite during cooling (Ar_3_) was 752 °C. In comparison, the temperature at the onset of austenite to pearlite transformation during cooling (Ar_1_) was 639 °C. Taking into account the earlier rolling stages [30] and analyzing the cooling curve (Figure 1), it can be concluded that both the rolling and cooling of steel in this variant took place in a single-phase range. The proportions of ferrite and pearlite under these thermoplastic processing conditions were 84.4% and 15.6%, respectively. The average hardness value was 186 *HV*. The mechanical properties of the material formed under these conditions were 370 MPa yield strength and 560 MPa ultimate tensile strength, while the plasticity reserve amounted to 0.661. The increase in hardness and analyzed mechanical properties in this variant compared with variant V1 (despite the higher end-of-rolling temperature) may be due, among other factors, to the higher cooling rate.

Numerical modeling of thermoplastic processing of a 5.5 mm diameter wire rod made of 20MnB4 steel, according to variant V3, led to the following results: the temperature of the beginning of the transformation of austenite to ferrite during cooling (Ar_3_) was 742 °C, while the temperature at the beginning of the transformation of austenite to pearlite during cooling (Ar_1_) was 631 °C. Comparing the Ar_3_ temperatures for variants V2 and V3, it can be noted that an increase in the cooling rate (from the same temperature) results in a decrease in the Ar_3_ and Ar_l_ transformations temperature, which is consistent with data published in, inter alia, [45]. Upon analyzing the earlier stages of the rolling process [30] and the cooling curve (Figure 1), it was determined that both rolling and final cooling of steel in this variant took place in the single-phase range. Under these conditions, the percentage ratios of ferrite and pearlite were 83.7% and 16.3%, respectively. The average hardness was 194 *HV*. The mechanical properties of the material formed under these conditions were yield strength—392 MPa and ultimate tensile strength—589 MPa. Plasticity reserve amounted to 0.666.

### 3.2. Physical Modelling Results—Multi-Sequence Non-Free Torsion—STD 812 Torsion Plastometer

As indicated, inter alia, in [1,33,34,35,36], in order to obtain a finished product with uniform fine-grained ferritic–pearlitic microstructure variations, the final stage of deformation (rolling in the RSM rolling block) should take place in the single-phase (austenitic) range, when its temperature is 30 ÷ 80 °C higher than the temperature at the beginning of the austenite transformation Ar_3_.

In order to check (verify) the microstructural state of the steel under investigation, physical modeling of the analyzed rolling process was carried out immediately prior to rolling in the RSM rolling block. This modeling was conducted in multi-sequence torsion tests according to the methodology described in detail in [32], using an STD 812 torsion plastometer. During the physical modeling, the tested steel was hardened immediately prior to the deformation stage, simulating a rolling wire rod in the RSM rolling block. It is not possible to carry out this type of research directly in the rolling line of the analyzed wire rod. Metallographic analysis and microhardness measurements of the resulting microstructure were then carried out.

Physical modeling was carried out in accordance with the temperature and deformation parameters shown in Table 2 and Table 3 and published, inter alia, in [30,32].

A schematic depiction of the physical modeling of wire rod rolling is shown in Figure 2.

In the beginning, the tested steel was heated to the temperature of 1165 °C, corresponding to the temperature in the leveling zone of a heating furnace (under industrial conditions). In order to obtain the same temperature over the entire volume of the sample working zone, the 20MnB4 steel was heated for 300 s. The next step was cooling for 30 s, to the temperature of 1086 °C, in simulation of the band cooling process during its transport from the furnace to the first stand of the rolling line. It was then deformed in 17 cycles, with deformation parameters shown in Table 2, replicating the rolling process in a continuous rolling mill. The subsequent stage consisted of accelerated cooling to the temperature of 851 °C, corresponding to the band temperature before the NTM rolling block of the continuous rolling mill. In the next physical modeling step, the rolling process in the NTM rolling block was simulated (Table 2), following which the accelerated cooling process between the NTM and RSM rolling blocks was modeled, according to the variant, down to 800 °C (variant V1) or to 850 °C (variants: V2 and V3) and then hardened at a cooling rate of approximately 130 °C/s in variant V1 or 146 °C/s in variants V2 and V3, using helium.

After physical modeling, samples of the material were subjected to metallographic testing. The samples were treated with ferric chloride for 90 s. In order to precisely identify the individual phases, additional microhardness measurements *HV* 0.01 (pressing force value 0.09807 N, time 10 s) were carried out.

Figure 3 shows examples of 20MnB4 steel microstructures after physical modeling.

Microstructure analysis of 20MnB4 steel samples after physical modeling of the 5.5 mm in diameter wire rod rolling process shows that in all analyzed technological variants, the material after hardening had a martensitic structure over its entire cross-section with some residual austenite (not exceeding 4%). In the case of the V1 variant, the average microhardness of the tested samples was approximately 369.11 *HV* 0.01. The average microhardness of the tested samples for V2 and V3 variants was approximately 416.96 *HV* 0.01. Since no other phases were detected in the tested samples, it can be concluded that, under real conditions (in all analyzed variants), the investigated steel was deformed in the RSM rolling block of the wire rod mill in the single-phase (austenitic) state.

### 3.3. Thermovision Investigation Results—ThermaCAM SC640 Thermovision Camera

One of the factors affecting the uniformity of the microstructure and mechanical properties of the wire rod along its length is the temperature distribution along the length of the rolled feedstock. This section presents the results of measurements of the surface temperature distribution along the length of the rolled material in several places of the rolling line, including at the exit of the heating furnace, before rolling stands No. 1 and 15, before the NTM (No-Twist Mill) rolling block, before the wire rod coil former, and at the entrance to the rolling conveyor in the STELMOR^®^ system. The testing was carried out using thermovision technology—a FLIR Systems ThermaCAM SC640 thermovision camera equipped with an uncooled detector [38]. Temperature measurements were recorded as measured video sequences, in which the areas of maximum recorded temperatures (marked by the presence of mill scale on the surface of the tested steel) were analyzed. ThermaCAM Researcher Professional software was used to process these data. Surface temperature investigations were preceded by the determination of the 20MnB4 steel emissivity within the temperature range under study (700 ÷ 1200 °C). A description of the methods for determining the emissivity can be found, inter alia, in [46,47]. Based on the results of these studies, it was determined that the emissivity of 20MnB4 steel in the analyzed temperature range varied from 0.80 to 0.85.

Examples of thermograms and a graph of the surface temperature distribution along the length of the rolled material in a number of places of the rolling line (for variants V2 and V3) are shown in Figure 4, Figure 5 and Figure 6.

Based on the measurements of the surface temperature distribution along the length of the rolled material (Figure 5), it was determined that the feedstock heated in the furnace was characterized by a high uniformity of the temperature distribution along its length. The average surface temperature of the feedstock at the exit of the heating furnace was approximately 1074 °C, whereas the maximum difference in surface temperature along the length of the feedstock after the heating process was about 50 °C (Figure 5). According to industry data, the temperature in the last zone of the heating furnace (the so-called leveling zone) was approximately 1165 °C. The error between the temperature measured with the thermovision camera and industrial data was 7.8% and was mainly caused by the large amount of mill scale on the surface of the processed material.

As a result of air cooling during the transport of the feedstock to the first rolling stand, its average surface temperature decreased to approximately 1060 °C (Figure 4). At the same time, due to the long rolling time in the first rolling stand and the different air-cooling times of the beginning and end of the 14-m-long feedstock, the difference in surface temperature distribution along its length increased to about 116 °C (Figure 5).

In the subsequent stages of the rolling process, due to successive thermoplastic processing operations, a decrease in cross-sectional area, an increase in rolling speed, accelerated cooling, the generation and conduction of heat resulting from plastic strain, and shorter intervals between deformations, it was found that the surface temperature difference along the length of the rolled band decreased again. The average surface temperature of the rolled material in front of rolling stand No. 15 was approximately 1065 °C. In contrast, the temperature difference along the length of the processed material at this point of the rolling line was approximately 66 °C (Figure 5). The average surface temperature of the material before being processed in the NTM rolling block was approximately 830 °C, while the temperature difference along the band at this point of the rolling line was approximately 34 °C (Figure 5). The average surface temperature of the band in front of the wire coil former was about 863 °C, while the temperature difference along its length at this point in the rolling line was about 48 °C (Figure 5).

An analysis of the surface temperature measurements of the rolled steel grade, shown in Figure 6, led to the conclusion that the surface temperature difference again increased along the length of the wire rod at the entry to the STELMOR^®^ roller conveyor. The increase in surface temperature along the length of the wire rod and its large variation at this point is due to the characteristic arrangement of the wire rod coils and the different heat transfer conditions in the individual areas of the roller conveyor. The temperature difference along the length of the wire rod at this point in the rolling line was as high as 177 °C (Figure 6).

### 3.4. Mechanical Properties Test Results and Upsetting Tests Results along Wire Rod Length—Zwick Z/100 Testing Machine

This section presents the results of tests of selected mechanical properties of the finished wire rod along its length (including their statistical analysis), produced according to technological variants V1–V3. The tests were carried out in a static tensile test according to the standard [39]. The gage length of the extensometer was 100 mm. Each time, a total of 20 samples were taken from the middle and end parts of the rolled band (Figure 7a,b). A general view of the 20MnB4 steel wire rod before and after the tensile test is shown in Figure 7c,d. Figure 8 shows examples of tensile curves of samples of 20MnB4 steel grade wire rod, produced according to technological variants V1–V3. Detailed results of the finished wire rod’s mechanical properties tests along its length produced according to technological variants V1–V3 are presented in Table 5.

A basic statistical analysis of mechanical properties along the length of the wire rod is presented below. Basic statistical indicators (arithmetic average, standard deviation, variation coefficient, median) were calculated.

In addition, graphs were made of, inter alia, the changes in the analyzed mechanical properties along the length of the wire rod, as well as graphs of normal distribution.

Based on data presented in Table 5 and Table 6, the average yield strength of the wire rod produced according to variant V1 was 302.88 MPa, while the average ultimate tensile strength was 513.17 MPa. In turn, the average value of unit elongation was approximately 22.30%. The average value of the relative reduction in area at fracture was ca. 68.87%. The average plasticity reserve of the wire rod was about 0.590.

An analysis of the mechanical properties of the wire rod produced according to variant V2 led to the conclusion that the average value of the yield strength of the wire rod was 359.30 MPa, which is approximately 19% higher than the average value of the yield strength of the wire rod produced according to variant V1. The average ultimate tensile strength of the wire rod produced according to variant V2 was similar to the average yield strength of the wire rod produced according to variant V1 and was 515.12 MPa. The average unit elongation of the wire rod produced according to variant V2 was approximately 23.02% and was ca. 3% higher than the average unit elongation of the wire rod produced according to variant V1. The average relative reduction in area at fracture of the wire rod produced according to variant V2 was 69.03% and was comparable with the corresponding value of the wire rod produced according to the V1 variant. The average plasticity reserve of the wire rod produced according to the V2 variant was approximately 0.698, about 18% higher than the average plasticity reserve of the wire rod produced according to variant V1.

Based on the analysis of results of the mechanical properties tests conducted on the wire rod produced according to variant V3, it was concluded that the average yield strength of the wire rod was 398.95 MPa, which is approximately 32% higher than the average yield strength of wire rod produced according to variant V1, and 11% higher than the average yield strength of wire rod produced according to variant V2. The mean value of the ultimate tensile strength of the wire rod produced according to variant V3 was 561.52 MPa, approximately 9% higher than the mean value of the ultimate tensile strength of the wire rod produced according to variants V1 and V2. The mean value of unit elongation of wire rods produced according to variant V3 was similar to the average value of unit elongation of wire rods produced according to variant V1, ca. 20.73%. This value was approximately 10% lower than the average unit elongation value of the wire rod produced according to the V2 variant. However, this did not adversely affect its capacity to be further subjected to the cold-forming process, which was confirmed using cold-upsetting tests with relative strain as high as 75%. The average value of the relative reduction in area at fracture of wire rod produced according to the V3 variant was approximately 72.75%, ca. 6% higher than the average relative reduction in area at fracture produced according to V1 variant and approximately 5% higher than the average value of the relative reduction in area at fracture in case of wire rod produced according to V2 variant. Whereas the average plasticity reserve of wire rod produced according to V3 variant was approximately 0.711, ca. 21% higher than the average plasticity reserve of wire rod produced according to V1 variant and ca. 2% higher than the average plasticity reserve of wire rod produced according to V2 variant.

Furthermore, upon analyzing data presented in Table 6, it was found that in the case of wire rods produced according to the V1 variant, the standard deviation values were 18.34 for yield strength, 10.08 for ultimate tensile strength, 1.44 for unit elongation and 1.03 for relative reduction in area at fracture. The variation coefficients of wire rods produced according to the V1 variant were 0.061 for yield strength, 0.02 for ultimate tensile strength, 0.065 for unit elongation, and 0.015 for relative reduction in area at fracture.

For wire rods produced according to the V2 variant, the standard deviation values were 7.03 for yield strength, 13.28 for ultimate tensile strength, 1.46 for unit elongation, and 1.25 for relative reduction in area at fracture. The variation coefficients in the case of wire rods produced according to the V2 variant were 0.02 for yield strength, 0.026 for ultimate tensile strength, 0.064 for unit elongation, and 0.018 for relative reduction in area at fracture.

The standard deviation values for wire rods produced according to the V3 variant were 9.50 for yield strength, 11.77 for ultimate tensile strength, 1.69 for unit elongation, and 1.3 for relative reduction in area at fracture. In contrast, the variation coefficients for wire rods produced according to the V3 variant were 0.024 for yield strength, 0.021 for ultimate tensile strength, 0.082 for unit elongation, and 0.018 for relative reduction in area at fracture.

Based on the analysis of data presented in Table 5 and Table 6, it is difficult to unambiguously assess the uniformity of the analyzed mechanical properties along the length of wire rods subjected to different thermoplastic processing conditions. Each of the three technological variants is characterized by a certain dispersion of the analyzed mechanical properties along the wire rod length. The values of the standard deviation and the variation coefficient depend on both the end-of-rolling temperature and the cooling rate after the rolling process. Therefore, additional graphs of the variation in the analyzed mechanical properties along the wire rod length and of normal distribution were compiled.

Data presented in Figure 9, Figure 10, Figure 11 and Figure 12 shows that each of the analyzed technological variants is characterized by some degree of nonuniformity of the mechanical properties along the length of 20MnB4 steel wire rod with a diameter of 5.5 mm, which, however, retains some stability. This nonuniformity is caused by a number of factors. One of the most important is the temperature distribution along the processed wire rod. The greatest nonuniformity of the temperature distribution along the wire rod was observed during the controlled cooling after the roller track process (Section 3.3). The nonuniformity of the temperature distribution causes, in turn, nonuniformity of the microstructure and, consequently, nonuniform properties of the finished wire rod.

An analysis of data shown in Figure 9a led to the conclusion that the highest yield strength value of wire rods produced according to the V1 variant was about 336 MPa, while the lowest was about 267 MPa. The highest yield strength value of wire rods produced according to the V2 variant was 376 MPa, and the lowest ca. 348 MPa. In contrast, the highest yield strength value of wire rods produced according to the V3 variant was about 418 MPa, and the lowest was around 381 MPa.

Based on data shown in Figure 9b, it was determined that the highest value of the ultimate tensile strength of the wire rod produced according to the V1 variant was about 524 MPa, and the lowest was around 490 MPa. The highest value of the ultimate tensile strength of the wire rod produced according to the V2 variant was 546 MPa, while the lowest was about 497 MPa. The highest value of ultimate tensile strength of the wire rod produced according to the V3 variant was about 588 MPa, and the lowest was around 541 MPa.

Based on data presented in Figure 9c, the highest unit elongation value of wire rods produced according to the V1 variant was 25.14%, and the lowest was 19.1%. The highest value of relative elongation of wire rods produced according to variant V2 was 24.89%, and the lowest was 20.61%. In turn, the highest unit elongation value of wire rods produced according to the V3 variant was 23.8%, and the lowest was 16.53%.

Based on data shown in Figure 9d, it was determined that the highest value of the relative reduction in area at fracture of wire rods produced according to the V1 variant was 70.46%, and the lowest was 67.2%. The relative reduction in area at fracture of wire rod produced according to the V2 variant was 71.8%, and the lowest was 67.02%. In turn, the highest value of the relative reduction in area at fracture of wire rod produced according to the V3 variant was 74.92%, and the lowest was 69.84%.

In order to determine whether the observed nonuniformity of mechanical properties along the wire rod is significant regarding the possibility of the tested steel undergoing further cold forming processing, additional cold upsetting tests were carried out for each technological variant according to the standard [41] as well as an assessment of the surface quality concerning the occurrence of any cracks. A minimum of three samples taken from the middle and end parts of the rolled material were tested each time. Figure 13 shows examples of samples after the cold upsetting process with a relative plastic strain of 50%, 67%, and 75%.

Upon analyzing the test results, it was found that there were no cracks, scratches, or other surface defects on any surfaces of the technological variants analyzed, even after applying a relative plastic strain of 75% (the sample’s hight indicator after cold upsetting 0.25).

In summary, the results of the tests regarding the mechanical properties of the finished wire rod along its length and of the upsetting test, it was found that for the studied thermoplastic processing parameters, there was no significant impact of the production process parameters on the quality of the finished steel product. This is partly due to the narrow rolling temperature range in the RSM block (800 °C and 850 °C). However, we have to keep in mind that the rolling temperature range in the RSM block covered by this study is typical for 20MnB4 steel. On the other hand, comparing the mechanical properties of the wire rod produced in the different technological variants, it was noted that the best complex of mechanical properties has the wire rod processed in the RSM rolling block at 850 °C and cooled after the rolling process on a roller conveyor at a rate of 10 °C/s (V3 technological variant). The wire rod produced in this way also has the required level of plasticity reserve, which enables further deformation of the studied steel grade in upsetting tests with a relative plastic strain of 75%.

### 3.5. Results of Metallographic and Hardness Tests of the Wire Rod along Its Length—Nikon Eclipse MA 200 Microscope, FM 700 Microhardness Tester

This section presents an analysis of the results of metallographic observations and measurements of the samples after the rolling process according to V1–V3 variants. A Nikon Eclipse MA-200 microscope with NIS-Elements software was used for the study [40]. In each case, metallographic observations and measurements of ferrite grain size (*D_α_*) (perpendicular secant method [48]) were carried out on longitudinal sections of samples taken from the middle and end part of the rolled wire rod at a total of 30 points (5 points on the wire rod radius at several places along the wire rod length). In addition, Vickers hardness measurements were taken with FutureTech’s FM 700 microhardness tester [40].

The aim of the testing was to determine the effect of the applied thermoplastic processing conditions on the microstructure and hardness of a 5.5 mm diameter wire rod made of 20MnB4 steel.

A diagram of the longitudinal section of the wire rod with the characteristic points plotted is shown in Figure 14. Detailed results of the ferrite grain size measurements (*D_α_*) and hardness along the length of 20MnB4 steel wire rod with a diameter of 5.5 mm, produced according to technological variants V1–V3, are shown in Table 7. Photographs of the microstructure (in longitudinal section) of a 5.5 mm diameter wire rod made from 20MnB4 steel produced according to process variants V1–V3 are shown in Figure 15, Figure 16 and Figure 17.

Based on the analysis of the test results shown in Table 7, the average *D_α_*—ferrite grain size of the wire rod produced according to the V1 variant was 16.21 μm. The average ferrite grain size of the wire rod produced according to the V2 variant was approximately 39% smaller than that of the wire rod produced according to the V1 variant and was 9.86 μm. In turn, the average ferrite grain size of the wire rod produced according to the V3 variant was 8.07 μm and was approximately 50% smaller than the wire rod produced according to the V1 variant and around 18% smaller than the wire rod produced according to V2 variant. The greater fragmentation in the average ferrite grain observed in the case of the wire rod produced according to V2 and V3 variants was mainly due to the higher cooling rate on the roller track, which prevented grain growth. When analyzing the distribution of ferrite grain size along the radius of the wire rod, it was found that the largest ferrite grains were in the axis of the wire rod, while the smallest was observed at its surface. This was due, among other things, to the temperature distribution over the cross-section of the wire rod and the heat transfer conditions (higher temperature in the axis of the wire rod than at its surface).

Upon analyzing the results of the hardness measurements, it was determined that the average hardness of the wire rod produced according to the V1 variant was 177.20 *HV*. The average hardness of the wire rod produced according to the V2 variant was 182.18 *HV*, which is ca. 3% higher than the average hardness value of the wire rod produced according to the V1 variant. The average hardness of the wire rod produced according to the V3 variant was 198.17 *HV*, around 12% higher than the average hardness value of the wire rod produced according to variant V1 and ca. 9% higher than the average hardness value of the wire rod produced according to variant V2. An analysis of the hardness distribution along the radius of the wire rod led to the conclusion that it increases as the size of the ferrite grain becomes smaller.

Metallographic analysis showed that the wire rod produced according to the V1 variant (Figure 15) had a nonuniform coarse-grained ferritic–pearlitic microstructure in the form of alternating bands of ferrite and pearlite. The observed banding was mainly due to the slow cooling of the wire rod after the rolling process. The average ferrite grain size ranged from 12.64 μm to 19.04 μm. Upon analyzing the microstructure of the wire rod produced according to the V2 variant (Figure 16), it was determined that increasing the cooling rate (despite the higher end-of-rolling temperature) resulted in favorable fragmentation and greater uniformity of the microstructure and significantly reduced its banding. The average ferrite grain size of the wire rod produced according to the V2 variant ranged from 4.66 μm to 13.78 μm. As depicted in Figure 17, increasing the cooling rate resulted in even greater fragmentation and uniformity of the ferrite grain size. The microstructure of the wire rod produced according to the V3 variant was also characterized by the lowest banding. The average ferrite grain size, in this case, ranged from 4.01 μm to 10.27 μm.

## 4. Conclusions

Based on the theoretical and experimental results presented in this study, the following conclusions were drawn:Based on the analysis of the microstructure of 20MnB4 steel samples after physical modeling of the wire rod rolling process, it was determined that in all analyzed technological variants, the material after hardening had a martensitic structure throughout its mass with some residual autenite (not exceeding 4%). Since no other phases were identified in the test samples, it can be concluded that under real conditions (in all the analyzed variants), the tested steel was deformed in the RSM rolling block in a single-phase (austenitic) state.Statistical processing of the obtained test results and analysis of the normal distributions of the technological variants of the wire rod rolling process analyzed in this study showed some nonuniformity of the analyzed mechanical properties along the length of the finished product.One of the factors causing the nonuniformity of mechanical properties along the wire rod length is the different coil temperatures during the cooling of the wire rod, resulting from their characteristic arrangement on the roller conveyor—spiral.For the analyzed thermoplastic processing parameters, increasing the rolling temperature in the RSM block from 800 °C (V1) to 850 °C (V2) and increasing the cooling rate to 10 °C/s (V3) resulted in an improved combination of the analyzed mechanical properties.Under the analyzed thermoplastic processing conditions, wire rods produced according to technological variant V3 showed the lowest microstructure variation.Increasing the cooling rate in the STELMOR^®^ line in the V3 variant (despite the higher temperature of the rolled material in the RSM rolling block) resulted in greater fragmentation in the microstructure on the longitudinal section of the finished product by about 18% compared with the V2 variant and by about 50% compared with the V1 variant.In the studied range of thermoplastic processing parameters, the observed nonuniformity of mechanical properties along the length of the 5.5 mm diameter wire rod of 20MnB4 steel does not adversely affect the capacity for further cold-forming, which has been confirmed by upsetting tests with a relative plastic strain of up to 75%.

## Figures and Tables

**Figure 1 materials-17-00905-f001:**
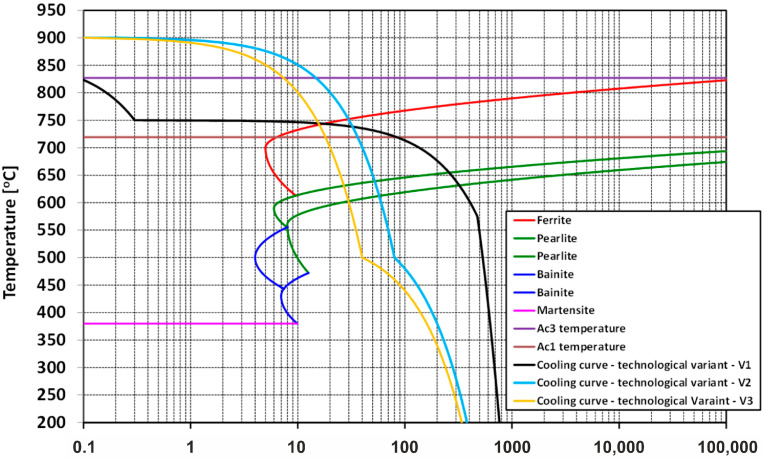
Part of the DTTT graph depicting investigated thermoplastic processing conditions for a 5.5 mm diameter wire rod of 20MnB4 steel.

**Figure 2 materials-17-00905-f002:**
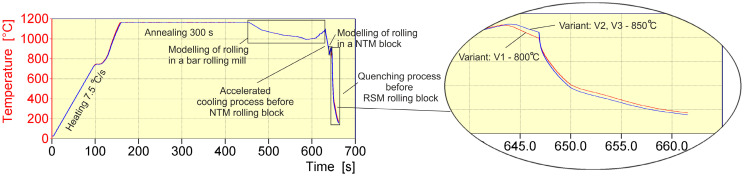
Scheme of physical modeling of rolling of 5.5 mm diameter wire rod of 20MnB4 steel grade.

**Figure 3 materials-17-00905-f003:**
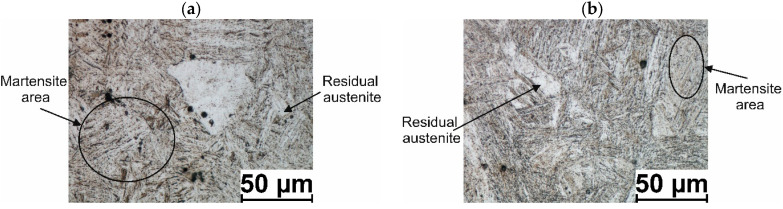
Example microstructure of 20MnB4 steel after physical modeling of wire rod rolling process: (**a**) technological variant—V1, (**b**) technological variant—V2, V3.

**Figure 4 materials-17-00905-f004:**
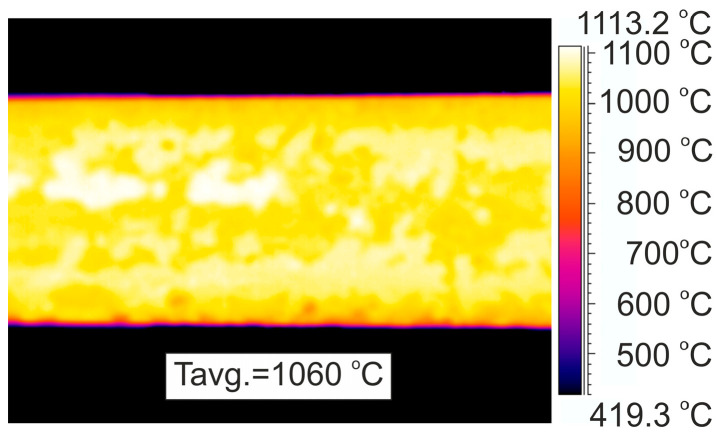
Example thermogram of temperature distribution on the surface of 20MnB4 steel grade band before rolling stand No. 1.

**Figure 5 materials-17-00905-f005:**
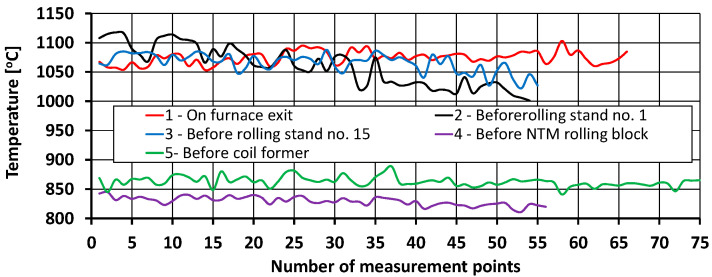
Diagram of surface temperature distribution along the length of the rolled material at various places in the rolling mill.

**Figure 6 materials-17-00905-f006:**
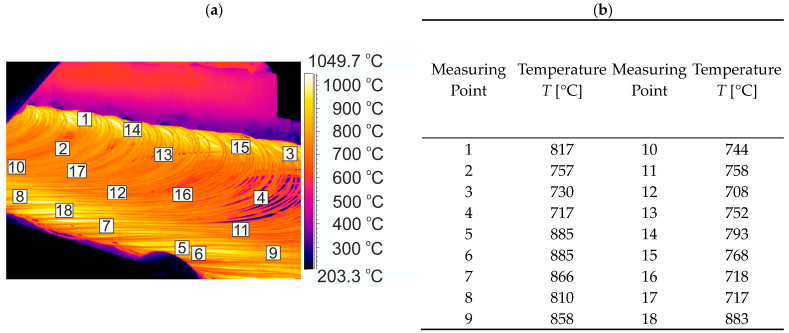
Example of temperature distribution on the surface of 20MnB4 steel wire rod at the entry to the STELMOR^®^ roller conveyor: (**a**) thermogram, (**b**) temperature value at measuring points.

**Figure 7 materials-17-00905-f007:**
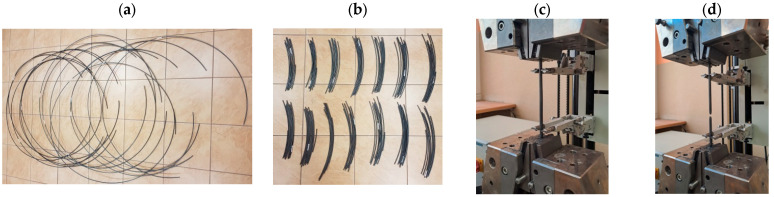
Mechanical properties tests along the length of 5.5 mm diameter wire rod of 20MnB4 steel grade: (**a**) wire rod coils, (**b**) samples for mechanical properties testing, (**c**) example sample before tensile test, (**d**) example sample after tensile test.

**Figure 8 materials-17-00905-f008:**
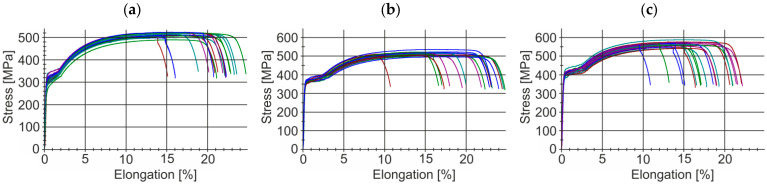
Examples of tensile curves of 20MnB4 steel grade wire rod samples: (**a**) technological variant—V1, (**b**) technological variant—V2, (**c**) technological variant—V3.

**Figure 9 materials-17-00905-f009:**
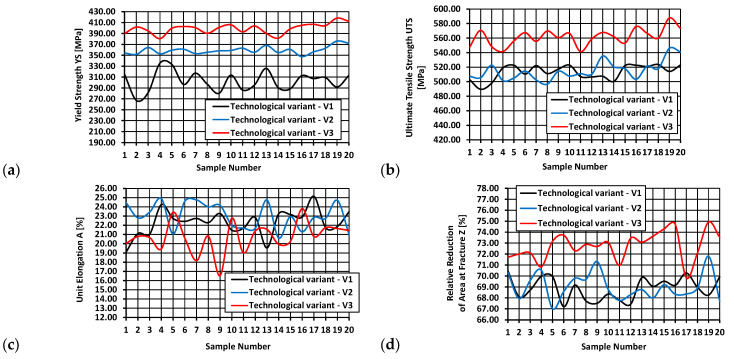
The course of changes in selected mechanical properties along the length of 5.5 mm diameter wire rod of 20MnB4 steel grade—technological variants V1–V3: (**a**) Yield Strength (YS), (**b**) Ultimate Tensile Strength (UTS), (**c**) Unit Elongation (A), (**d**) Relative Reduction in Area at Fracture (Z).

**Figure 10 materials-17-00905-f010:**
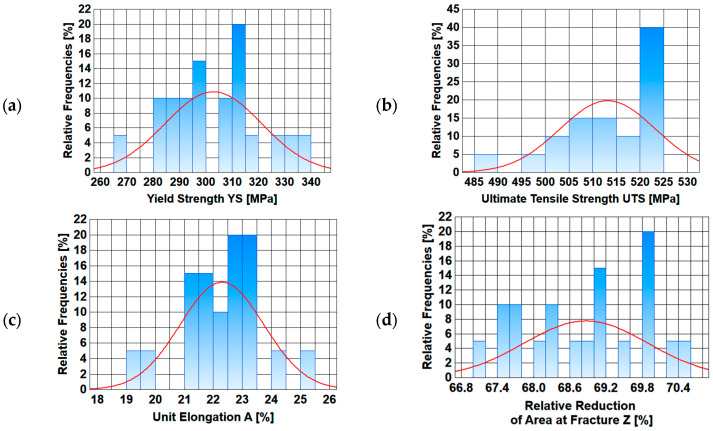
Normal distribution diagrams of selected mechanical properties along the length of a 5.5 mm diameter wire rod of 20MnB4 steel grade—technological variant V1: (**a**) Yield Strength (YS), (**b**) Ultimate Tensile Strength (UTS), (**c**) Unit Elongation (A), (**d**) Relative Reduction in Area at Fracture (Z).

**Figure 11 materials-17-00905-f011:**
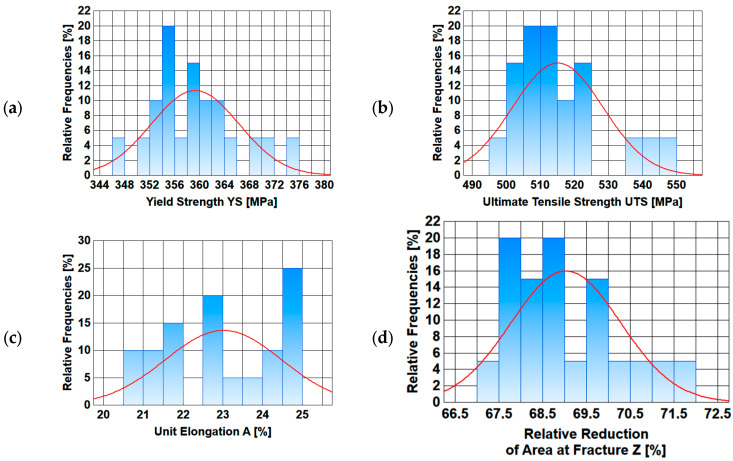
Normal distribution diagrams of selected mechanical properties along the length of a 5.5 mm diameter wire rod of 20MnB4 steel grade—technological variant V2: (**a**) Yield Strength (YS), (**b**) Ultimate Tensile Strength (UTS), (**c**) Unit Elongation (A), (**d**) Relative Reduction in Area at Fracture (Z).

**Figure 12 materials-17-00905-f012:**
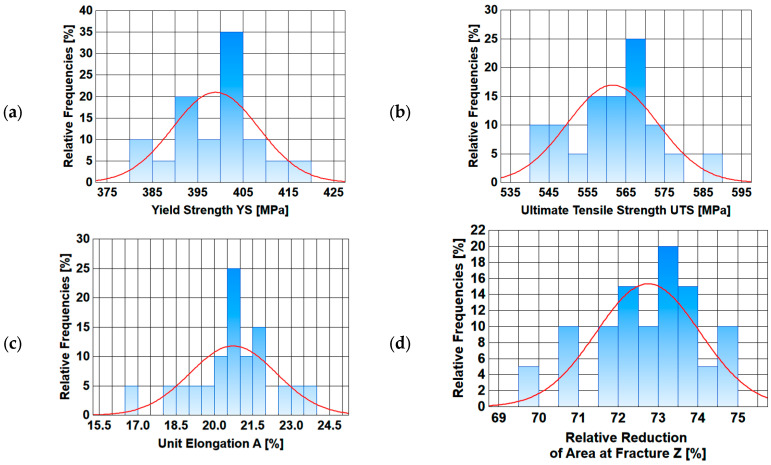
Normal distribution diagrams of selected mechanical properties along the length of a 5.5 mm diameter wire rod of 20MnB4 steel grade—technological variant V3: (**a**) Yield Strength (YS), (**b**) Ultimate Tensile Strength (UTS), (**c**) Unit Elongation (A), (**d**) Relative Reduction in Area at Fracture (Z).

**Figure 13 materials-17-00905-f013:**
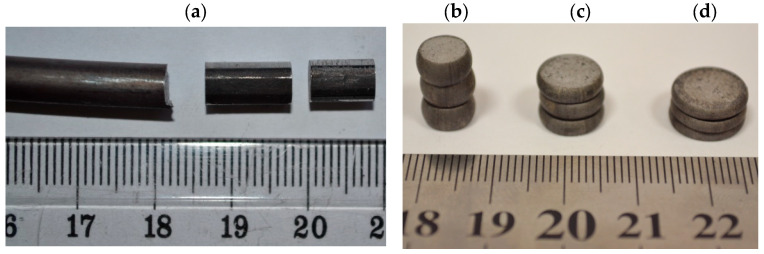
General view of wire rod of 20MnB4 steel grade samples for cold upsetting tests: (**a**) method of samples preparation from the finished product for cold upsetting tests, (**b**) view of samples after cold upsetting test (wire rod after the rolling process according to technological variant V2) with relative plastic strain value: (**b**) 50%, (**c**) 67%, (**d**) 75%.

**Figure 14 materials-17-00905-f014:**
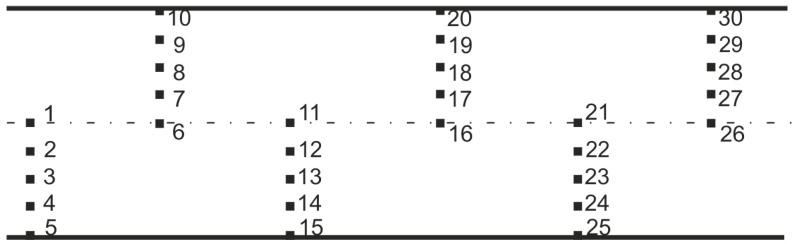
Scheme of a longitudinal section of wire rod with marked characteristic points.

**Figure 15 materials-17-00905-f015:**
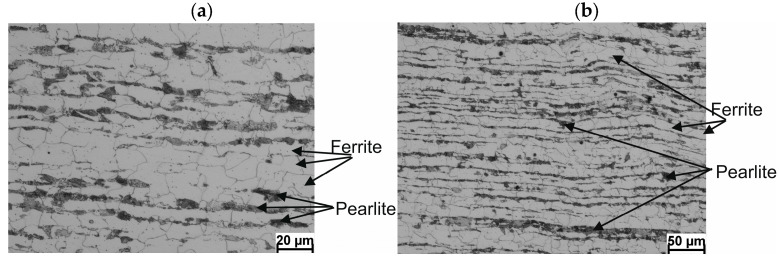
Example of the microstructure of a 5.5 mm diameter wire rod of 20MnB4 steel grade—longitudinal section—technological variant V1: (**a**) magnification 500×, (**b**) magnification 200×.

**Figure 16 materials-17-00905-f016:**
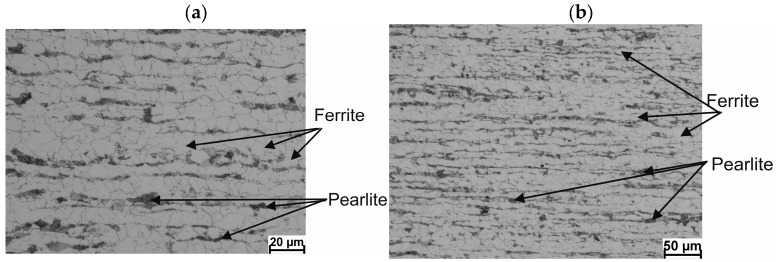
Example of the microstructure of a 5.5 mm diameter wire rod of 20MnB4 steel grade—longitudinal section—technological variant V2: (**a**) magnification 500×, (**b**) magnification 200×.

**Figure 17 materials-17-00905-f017:**
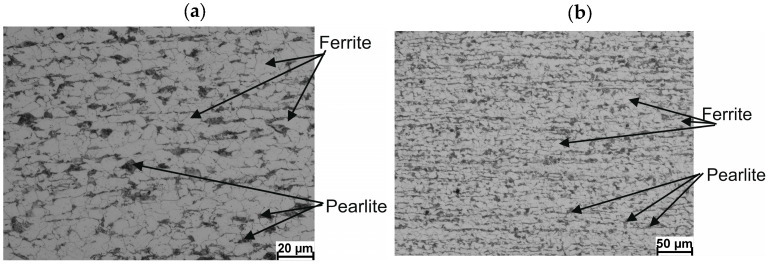
Example of the microstructure of a 5.5 mm diameter wire rod of 20MnB4 steel grade—longitudinal section—technological variant V3: (**a**) magnification 500×, (**b**) magnification 200×.

**Table 1 materials-17-00905-t001:** Chemical composition of 20MnB4 steel grade [29].

Steel Grade	Steel Number	Melt Analysis, mass%						
20MnB4	1.5525	C	Si	Mn	P_max_, S_max_	Cr	Cu_max_	B
		0.18 ÷ 0.23	≤0.30	0.90 ÷ 1.20	0.025	≤0.30	0.25	0.0008 ÷ 0.005

**Table 2 materials-17-00905-t002:** Parameters of 5.5 mm diameter wire rod rolling process of 20MnB4 steel grade ^1^.

Pass No.	Temperature T [°C]	Strain ε [−]	Strain Rate $\dot{\varepsilon }$ [s^−1^]	Break Time after Deformation t [s] ^2^	Pass No.	Temperature T [°C]		Strain ε [−]	Strain Rate $\dot{\varepsilon }$ [s^−1^]	Break Time after Deformation t [s] ^2^
	Continuous rolling mill					NTM block of wire rod rolling mill				
1	1086	0.18	0.16	26.47	18	851		0.49	156.02	0.091
2	1057	0.39	0.35	19.89	19	860		0.51	171.25	0.074
3	1037	0.28	0.39	29.98	20	867		0.56	276.33	0.058
4	1023	0.59	0.96	11.33	21	883		0.54	303.93	0.048
5	1010	0.46	1.15	8.91	22	892		0.56	477.46	0.037
6	995	0.50	2.02	6.13	23	908		0.53	584.28	0.032
7	997	0.45	2.45	11.65	24	918		0.62	991.51	0.024
8	1005	0.48	4.71	3.35	25	941		0.57	1042.10	0.020
9	1009	0.44	5.57	2.63	26	956		0.62	1753.46	0.015
10	1022	0.54	10.39	1.85	27	982		0.56	1809.67	0.82
11	1030	0.48	12.07	3.09		V1	V2, V3	RSM block of wire rod rolling mill		
12	1049	0.50	20.53	2.28	28	796	845	0.53	2368.05	0.012
13	1052	0.51	24.74	3.18	29	831	873	0.48	2275.43	0.007
14	1069	0.50	46.34	1.35	30	850	894	0.13	1853.11	0.004
15	1072	0.41	47.13	1.11	31	853	895	0.10	1680.68	
16	1087	0.51	79.93	0.90						
17	1091	0.31	70.63	8.52						

^1^ Table based on data published at works [30,31,32], ^2^ transport time of band between successive rolling stands.

**Table 3 materials-17-00905-t003:** Parameters of the controlled cooling process after rolling of 5.5 mm diameter wire rod of 20MnB4 steel grade ^1^.

Temperature before RSM Rolling Block T [°C]		Cooling Method after Rolling Process ^2^		
		Stage No. 1		
		Desired Temperature Value T [°C]	Cooling Time t, s	Cooling Rate C_r_, °C/s
800	(V1)	575	475	0,4
850	(V2)	500	70	5
850	(V3)	500	35	10

^1^ Table based on data published in [30], ^2^ In the second cooling stage, the studied steel was cooled to 200 °C at a rate of 1 °C/s.

**Table 4 materials-17-00905-t004:** Shares of microstructure components, hardness, selected mechanical properties, and characteristic temperature for the conditions of thermoplastic processing of a 5.5 mm diameter wire rod of 20MnB4 steel grade.

Temperature before RSM Rolling Block T [°C]		Contribution of Microstructure Components [%]		Vickers Hardness [*HV*]	Yield Strength YS [MPa]	Ultimate Tensile Strength UTS [MPa]	Plasticity Reserve YS/UTS	Characteristic Temperature			
		Ferrite	Pearlite					Ac_3_	Ac_1_	Ar_3_	Ar_1_
800	(V1)	94.5	5.5	173	319	516	0.618	827	719	744	655
850	(V2)	84.4	15.6	186	370	560	0.661			752	639
850	(V3)	83.7	16.3	194	392	589	0.666			742	631

where: Ac_1_—temperature of the beginning of the transformation of ferrite into austenite during heating, Ac_3_—temperature of the end of the transformation of ferrite into austenite during heating, Ar_3_—temperature of the beginning of austenite transformation into ferrite during cooling, Ar_1_—temperature of the beginning of the transformation of austenite into pearlite during cooling.

**Table 5 materials-17-00905-t005:** Mechanical properties along the length of 5.5 mm diameter wire rod of 20MnB4 steel grade.

Sample Number	Technological Variant—V1					Technological Variant—V2					Technological Variant—V3				
	Yield Strength YS [MPa]	Ultimate Tensile Strength UTS [MPa]	Unit Elongation A [%]	Relative Reduction in Area at Fracture Z [%]	Plasticity Reserve YS/UTS	Yield Strength YS [MPa]	Ultimate Tensile Strength UTS [MPa]	Unit Elongation A [%]	Relative Reduction in Area at Fracture Z [%]	Plasticity Reserve YS/UTS	Yield Strength YS [MPa]	Ultimate Tensile Strength UTS [MPa]	Unit Elongation A [%]	Relative Reduction in Area at Fracture Z [%]	Plasticity Reserve YS/UTS
1	2	3	4	5	6	7	8	9	10	11	12	13	14	15	16
1	314.93	502.74	19.10	70.46	0.626	354.12	507.40	24.43	70.32	0.698	389.76	547.35	20.02	71.71	0.712
2	267.18	489.56	21.10	68.10	0.546	351.74	505.57	22.80	67.94	0.696	401.69	570.55	20.78	72.00	0.704
3	281.48	498.20	21.04	68.72	0.565	364.18	522.59	23.42	69.60	0.697	394.54	548.26	20.66	72.11	0.720
4	335.63	518.86	24.24	69.97	0.647	352.72	501.56	24.89	70.51	0.703	380.49	541.51	19.44	70.84	0.703
5	333.23	522.58	22.73	69.95	0.638	359.33	505.09	21.06	67.02	0.711	399.55	556.18	23.40	73.16	0.718
6	296.57	510.49	22.42	67.20	0.581	361.14	514.56	24.61	68.58	0.702	402.74	567.44	20.59	73.72	0.710
7	317.25	522.13	22.74	69.17	0.608	353.16	502.45	24.74	69.78	0.703	400.64	555.63	18.16	72.31	0.721
8	297.08	511.19	22.28	67.72	0.581	355.45	496.54	24.00	69.69	0.716	390.24	569.73	20.83	72.88	0.685
9	280.09	517.29	23.27	67.51	0.541	358.10	514.39	24.16	71.32	0.696	400.97	560.57	16.53	72.69	0.715
10	313.50	522.80	21.47	68.35	0.600	358.80	507.77	22.00	68.76	0.707	405.96	566.54	22.75	73.07	0.717
11	286.24	507.12	21.68	67.78	0.564	362.95	510.95	21.69	67.77	0.710	392.96	541.29	19.00	70.96	0.726
12	295.42	506.49	22.85	67.42	0.583	355.48	510.26	21.66	68.29	0.697	403.71	558.66	21.37	73.44	0.723
13	326.01	507.31	19.54	69.84	0.643	368.36	535.31	24.74	68.76	0.688	390.20	567.80	21.55	73.08	0.687
14	291.13	500.29	23.24	69.05	0.582	355.06	520.59	20.61	67.99	0.682	381.31	562.00	19.93	73.62	0.678
15	287.95	522.09	23.16	69.52	0.552	360.88	517.94	22.94	69.20	0.697	398.01	553.20	20.25	74.29	0.719
16	312.53	522.65	22.88	69.15	0.598	347.54	503.24	21.29	68.34	0.691	404.93	575.94	23.80	74.73	0.703
17	307.24	520.99	25.14	70.27	0.590	356.16	521.38	22.84	68.36	0.683	406.82	566.57	20.78	69.84	0.718
18	309.41	523.65	21.74	68.97	0.591	362.79	517.90	22.75	68.89	0.701	404.21	560.22	21.75	72.10	0.722
19	291.87	514.10	21.86	68.26	0.568	376.00	546.00	24.70	71.80	0.689	418.14	587.66	21.64	74.92	0.712
20	312.91	522.95	23.47	69.93	0.598	372.00	541.00	21.00	67.70	0.688	412.18	573.28	21.43	73.52	0.719
Average value	302.88	513.17	22.30	68.87	0.590	359.30	515.12	23.02	69.03	0.698	398.95	561.52	20.73	72.75	0.711

**Table 6 materials-17-00905-t006:** Basic statistical indicators of 5.5 mm diameter wire rod of 20MnB4 steel grade produced according to technological variants V1–V3.

Statistical Parameter	Technological Variant—V1				Technological Variant—V2				Technological Variant—V3			
	Yield Strength YS [MPa]	Ultimate Tensile Strength UTS [MPa]	Unit Elongation A [%]	Relative Reduction in Area at Fracture Z [%]	Yield Strength YS [MPa]	Ultimate Tensile Strength UTS [MPa]	Unit Elongation A [%]	Relative Reduction in Area at Fracture Z [%]	Yield Strength YS [MPa]	Ultimate Tensile Strength UTS [MPa]	Unit Elongation A [%]	Relative Reduction in Area at Fracture Z [%]
1	2	3	4	5	6	7	8	9	10	11	12	13
Arithmetic average	302.88	513.17	22.30	68.87	359.30	515.12	23.02	69.03	398.95	561.52	20.73	72.75
Standard deviation	18.34	10.08	1.44	1.03	7.03	13.28	1.46	1.25	9.50	11.77	1.69	1.30
Coefficient of variation	0.061	0.020	0.065	0.015	0.020	0.026	0.064	0.018	0.024	0.021	0.082	0.018
Median	302.16	515.70	22.58	69.01	358.45	512.67	22.89	68.76	400.81	561.29	20.78	72.98

**Table 7 materials-17-00905-t007:** Measurements of ferrite grain size and hardness along the length of a 5.5 mm diameter wire rod of 20MnB4 steel grade were produced according to technological variants V1–V3.

Technological Variant—V1			Technological Variant—V2			Technological Variant—V3		
Measurement Location (According to the Figure 14)	Average Ferrite Grain Size *D_α_* [μm]	Average Hardness Value [*HV*]	Measurement Location (According to the Figure 14)	Average Ferrite Grain Size *D_α_* [μm]	Average Hardness Value [*HV*]	Measurement Location (According to the Figure 14)	Average Ferrite Grain Size *D_α_* [μm]	Average Hardness Value [*HV*]
1	16.82	175.90	1	12.24	174.40	1	8.95	193.50
2	16.81	178.30	2	11.40	175.90	2	7.58	197.50
3	15.18	178.70	3	9.13	183.15	3	7.57	197.75
4	13.96	181.15	4	8.80	186.60	4	6.15	201.50
5	12.73	184.25	5	8.77	194.35	5	5.51	216.90
6	17.39	174.45	6	12.38	171.05	6	10.27	189.30
7	15.96	176.00	7	11.97	172.25	7	8.71	194.70
8	14.30	182.20	8	9.63	176.40	8	8.43	196.05
9	13.88	182.30	9	8.72	188.00	9	8.40	197.00
10	13.29	188.10	10	8.46	195.80	10	7.13	201.45
11	18.36	172.60	11	13.78	169.85	11	9.27	192.05
12	15.35	179.85	12	11.16	176.13	12	8.64	196.15
13	15.00	180.00	13	9.34	177.83	13	7.69	197.75
14	14.03	181.10	14	9.09	178.55	14	7.24	198.55
15	12.64	184.90	15	4.92	201.40	15	6.53	204.75
16	18.57	169.85	16	11.46	176.70	16	9.78	190.55
17	18.41	170.50	17	9.99	178.15	17	9.78	191.90
18	17.95	173.25	18	9.27	181.55	18	9.68	192.60
19	16.79	176.15	19	9.20	183.10	19	9.18	196.45
20	16.75	176.90	20	8.91	192.15	20	9.00	198.00
21	18.29	171.30	21	11.12	180.70	21	10.08	190.45
22	17.91	173.75	22	10.50	181.15	22	9.76	192.05
23	16.61	173.95	23	9.05	187.10	23	7.22	199.35
24	16.19	178.45	24	8.61	189.10	24	7.14	199.45
25	15.86	182.40	25	4.66	210.77	25	4.01	222.65
26	19.04	168.80	26	11.69	172.90	26	9.41	192.75
27	18.31	172.55	27	11.08	174.75	27	9.21	193.05
28	16.94	173.50	28	10.64	176.30	28	7.16	199.50
29	16.89	174.25	29	10.39	178.90	29	6.87	200.00
30	15.99	180.45	30	9.56	180.50	30	5.79	211.35
Average value	16.21	177.20	Average value	9.86	182.18	Average value	8.07	198.17

## Data Availability

Data are contained within the article.

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
