# Peer review of "Analysis of the Uniformity of Mechanical Properties along the Length of Wire Rod Designed for Further Cold Plastic Working Processes for Selected Parameters of Thermoplastic Processing"

_materials, 2024, doi:10.3390/ma17040905_

Round 1

Reviewer 1 Report

Comments and Suggestions for Authors

The paper is well written, the text is clear and easy to read.
The conclusions are in line with the evidence and arguments presented.
Adequate methods were used, the achieved results are adequately presented with minor reservations.
Preparing of samples is very good described.
The numerical model was well designed.
Metallographic analysis is done at a very good level.
The results of the tensile test in fig.9 are not cleary determined –  description of samples is missing.  Indicate the gage length of the extensometer.
Upsetting tests were not necessarily needed.

Author Response

Response to Reviewer 1 Comments (Round 1)

Dear Reviewer.

Thank You very much for Your review. I am glad that the article resulted in positive feedback. Thank You very much also for Your comments.

Below You can find answers for Your comment – taking into account also others Reviewers comments and suggestions.

Point 1: The results of the tensile test in fig.9 are not cleary determined –  description of samples is missing.  Indicate the gage length of the extensometer.

Response 1: Thank You very much for this comment. You have right that description of samples is missing but these graphs is only graphical presentation of the obtained results. Mechanical properties for each sample are presented in table 7. Adding a legend with sample markings in Fig. 9 will reduce its readability. I can't increase the size of the fig. 9 because another reviewer suggests that the article is too long and suggests reducing it. Information about the measurement length of the accurate extensometer has been added. Details in the new version of the article file.

Point 2: Upsetting tests were not necessarily needed.

Response 2: Thank You very much for this suggestion. I made these tests to check if the observed heterogeneity of mechanical properties along the wire rod had influence for further cold plastic working processes. In was difficult for me to answer clearly for this question based on the results of mechanical properties tests or based on statistical analysis. That was the reason why I did this research. Moreover cold upsetting tests is base investigation for this grade of steel to investigate its ability for further cold plastic working processes.

Reviewer 2 Report

Comments and Suggestions for Authors

Comments and Suggestions are provided in the attached PDF.

Comments on the Quality of English Language

The author is not an English native speaker. Therefore, there several parts of the manuscript need revision (by an expert) in terms of English phrasing. For example, it is noted that “on the basis of” and “it was concluded” are repeated too many times along the text.

Author Response

Response to Reviewer 2 Comments (Round 1)

Dear Reviewer

Thank You very much for Your deep review.

Issues related to the rolling of long products, in particular bars and wire rods, are an important part of my research area. The main purpose of the paper [6] was influence of the non-uniform temperature distribution on the metallic charge length only on the energy and force parameters. Moreover this paper concerns round bars rolling process.

In the technical literature on the rolling process, it is possible to find papers dealing with the influence of the production process parameters on the quality of the finished product.

However, there are few papers that analyse the quality of the finished rolled product in terms of uniformity of microstructure and mechanical properties along its length, especially with regard to wire rod.

For this reason the undertaken research topic, that is the analysis of the uniformity of mechanical properties along the length of cold upsetting steel wire rod for selected thermoplastic processing parameters, is, in my opinion, valid. This is an important issue, given that wire rod from 20MnB4 steel is an input material for further cold plastic processing, e.g. for the drawing industry.

But thank you very much for noticing that this is a continuation of my previous research.

I was interested in it if the parameters of analysed rolling and cooling process (especially - temperature  and cooling rate) have influence of microstructure and mechanical properties (uniformity) on the length of the wire rod.

Below You can find my answers for Your suggestion and comments

Point 1: To make this paper more attractive, especially for those readers that are less familiar with the topic, explain/define/clarify what do you mean by “cold upsetting”.

Response 1: Cold upsetting steel wire rod – this expression means wire rod (as the finished product of the rolling process) which is then used as an input product for further processing by cold plastic working methods, for example in drawing processes  or for the production of nails (this particular grade of steel). It means wire rod for further cold plastic working processes. It is also a matter of translation. This was also checked in technical and industry dictionaries. Sometimes you can also find the term “cold-heading” (instead of cold upsetting). I was thinking which one will be better. I chose the term “cold upsetting” because I used this terminology in the past in my previous papers also. I modified a title of the manuscript. I hope that now it will be more understable. Modified title of the paper is: “Analysis of the uniformity of mechanical properties along the length of wire rod designed for further cold plastic working processes for selected parameters of thermoplastic processing”.

Moreover I add an additional explanation in the abstract and in the introduction (last paragraph) to make it more understandable. “Uniformity of mechanical properties along the length of wire rod designed for further cold plastic working processes is important problem. This is an important issue, given that wire rod from 20MnB4 steel is an input material for further cold plastic working processes, e.g. for the drawing processes or for the production of nails”.

You can find these changes in new version of my paper.

Point 2: At the Abstract (line 14): do not use "unit elongation" because this terminology is dubious.

Response 2: Thank You very much for this suggestion. I changed this terminology in the abstract and I used just “elongation”.

Point 3: Line 20: What do you mean by "stability"? Please clarify.

Response 3: In this particular case, when I used the expression stability, I meant that there is some variation in mechanical properties along the length of the wire rod. This is confirmed by obtained results- especially statistical analysis and normal distribution charts and graphs of course of changes in selected mechanical properties along the length of 5.5 mm diameter wire rod. But on the other hand the observed nonuniformity of mechanical properties along the length of the 5.5 mm diameter wire rod of 20MnB4 steel does not adversely affect the capacity for further cold-forming, which has been confirmed by upsetting tests with a relative plastic strain of up to 75%.

Point 4: The author is not an English native speaker. Therefore, there several parts of the manuscript need revision (by an expert) in terms of English phrasing. For example, it is noted that "on the basis of" and "it was concluded" are repeated too many times along the text.

Response 4: You're right, I'm not a native speaker. English is not my native language. Taking into account the specific industry terminology, the translation of the entire article was made by a company professionally engaged in translating scientific articles. Below I present the data of translation agency. Additionally, this information was sent to the Journal Editorial Office

VERBA-TEXT LLC Sp. K., Belgradzka 4/13 st., 02-793 Warszawa, Poland, www.verba-text.pl

Point 5: Table 2 and Table 3 have the same title (Parameters of 5.5 mm diameter wire rod rolling process of 20MnB4 steel grade in a continuous bar rolling mill [36-38]), and therefore they should be a single table.

Response 5: Table 2 and Table 3 haven’t the same title. Take a look please that Table 2 concerns parameters of rolling process in a continuous bar rolling mill while Table 3 concerns parameters of rolling process in NTM rolling block of wire rod rolling mill. But taking into account Your other suggestion that the manuscript is too long I decided to modify tables 2-3 into one table. You can find this table in the new version of my manuscript.

Point 6: The readers will ask: - What is "Break Time After Deformation" ?. Please explain.

Response 6: "Break Time After Deformation" it means the time during which the rolled band moves between successive rolling stands. I add an explanation of this term (table 2 in the new version of my manuscript)

Point 7: Lines 163 till 176: It is written "In variant one (VI) the temperature of the wire rod at the entrance to the RSM rolling block was approximately 800°C'. But the reader finds that values of V I in Table 4 reach 853°C. Reading the text and considering the temperature values shown in Table 4, similar temperature doubts arise for the variants V2 and V3. These issues need clarification.

Response 7: Thank You very much for this comment. By using the phrase “In variant one (V1) the temperature of the wire rod at the entrance to the RSM rolling block was approximately 800°C” I meant the temperature value at the entrance to the rolling pass. no 28 (first rolling stand of RSM rolling block). Similarly for the V2 and V3 variants. Because of high speed of rolling increasing of 20MnB4 steel grade temperature was observed in the RSM rolling block (rolling passes: 28-31). These results were have been confirmed in my previous work. The temperature value 853°C – it is temperature value of the rolled steel at the exit of RSM rolling block (rolling pass no. 31). Similarly for the V2 and V3 variants. The temperature increase is caused by heat generation due to high-speed strain rate. In this case, the temperature increase in each variant was approximately 50°C. According to Your suggestion I add a short information in a new version of my manuscript.

Point 8: Section 2. Materials and Methods: In a research article, the experiments section or "Materials and Methods" is an important section. All procedures followed by the authors should be clearly described. It is especially important that readers are informed about all details so that they can repeat the experiments if they are in possession of identical materials and equipment. Details about each relevant equipment used, as well as software, should be provided (namely: version or model, and manufacturer's or provider's name and country).

Response 8: I agree with your opinion. I thought I described it thoroughly. As You suggest  - the manuscript is to lenghtly at present. But according to Your suggestion, I added some additional information in the new version of the manuscript.

Point 9: Lines 271 & 272: The present title/caption of Figure 2 is dubious because the graph depicts only 3 cooling curves.

Response 9: I know that is not full DTTT diagram. Full DTTT diagram was published in below publications, among other things:

  1. Laber, K. Nowe Aspekty Wytwarzania Walcówki Ze Stali Do SpÄ™czania Na Zimno (New Aspects of Wire Rod Production from Steel for Cold Heading); Seria: Monografie nr 79 (Series: Monograph No. 79); Czestochowa University of Technology, Faculty of Production Engineering and Materials Technology Publishing House: CzÄ™stochowa, Poland, 2018; ISBN 978-83-63989-64-4. ISSN 2391-632X.

  1. Laber K.; Koczurkiewicz B. Determination of optimum conditions for the process of controlled cooling of rolled products with diameter 16.5 mm made of 20MnB4 steel. 24th International Conference on Metallurgy and Materials - METAL 2015, Brno, Czech Republic, June 3rd÷5th 2015, pp. 364÷370.

According to Your suggestion I modified caption of Figure 2 in the new version of the manuscript.

Point 10: Lines 298 & 299: Please explain here what you mean by “plasticity reserve".

Response 10: Plasticity reserve it is a YS (Yield Strength)/UTS (Ultimate Tensile Strength) ratio. I definied it shortly in table no. 6 and 7 (actual version of my manuscript). This parameter is used (also by manufacturers) to determine wire rod ability for further cold plastic processing, e.g. for the drawing industry.

Point 11: Lines 344 & 345 and title/caption of Figure 3: Reconsider modification of "thermoplastic processing during physical modelling of...", because it is dubious i.e. not clear.

Response 11: I modified this part of my manuscript according to Your suggestion. Modifications are presented in the new version file of the manuscript.

Point 12: Line 372, Figure 4: Once the rulers of 50 mm are depicted, it is does not make sense to mention 500x.

Response 12: Thank You very much for this comment. I deleted this information from the new version of manuscript.

Point 13: Section titled "Summary and conclusions" should be number 4 (instead of 5). The present text in this section is too long and contains some repetitions.

Response 13: Thank You very much for this comment. I changed number of this section. Moreover, I modified  this section according to Your suggestion.

Reviewer 3 Report

Comments and Suggestions for Authors

This is an interesting manuscript that requires some improvement
--Improve organization and novel aspects.
--Need to improve abstract and introduction to discuss the general problem and then present the specifics of this particular case.    The information is important but requires addition background into the actual problem before presenting the specifics.
--line 135 " with a final diameter of 5.5 mm" to supplement how big was the initial diameter

--Table 2,3,4, write formulas for calculating strain and strain rate

-- fig.4 a,b describe the content of the phases, into the picture

-- fig.16,17,18 describe the content of the phases, into the picture

--line 721 "the material after hardening had a martensitic structure" is not correct and what about residual austenite? how many percent?

--a lot of self-citation

Author Response

Response to Reviewer 3 Comments (Round 1)

Dear Reviewer

Thank You very much for Your review.

Below You can find my answers for Your suggestion and comments

Point 1: Improve organization and novel aspects.

Response 1: Thank you very much for this comment. According to Your suggestion I modified manuscript. Details You can find in the new version file.

In the technical literature on the rolling process, it is possible to find papers dealing with the influence of the production process parameters on the quality of the finished product.

However, there are few papers that analyse the quality of the finished rolled product in terms of uniformity of microstructure and mechanical properties along its length, especially with regard to wire rod.

For this reason the undertaken research topic, that is the analysis of the uniformity of mechanical properties along the length of cold upsetting steel wire rod for selected thermoplastic processing parameters, is, in my opinion, valid. This is an important issue, given that wire rod from 20MnB4 steel is an input material for further cold plastic processing, e.g. for the drawing industry or for the production process of nails.

Point 2: Need to improve abstract and introduction to discuss the general problem and then present the specifics of this particular case. The information is important but requires addition background into the actual problem before presenting the specifics.

Response 2: Thank You very much for this comment. According to Your suggestion I added some information at the end of the abstract and short information at the end of the introduction. But according to to the Reviewer 2 suggestion – at present  - my manuscript is too long. So I must to reduce my paper. I hope that these short information will be enough. Details in the new version of the manuscript.

Point 3: line 135 " with a final diameter of 5.5 mm" to supplement how big was the initial diameter

Response 3: The input material for the rolling process was an ingot from the continuous casting process with a square cross-section, side 160 mm and length 14000 mm. I added this information to the modified manuscript file, according to Your suggestion.

Point 4: Table 2,3,4, write formulas for calculating strain and strain rate

Response 4: Strain value and strain rate value were taken from my previous works. You can find the details in these works (among other things):

  1. Laber, K. Nowe Aspekty Wytwarzania Walcówki Ze Stali Do SpÄ™czania Na Zimno (New Aspects of Wire Rod Production from Steel for Cold Heading); Seria: Monografie nr 79 (Series: Monograph No. 79); Czestochowa University of Technology, Faculty of Production Engineering and Materials Technology Publishing House: CzÄ™stochowa, Poland, 2018; ISBN 978-83-63989-64-4. ISSN 2391-632X.
  2. Laber, K.; Knapiński, M. Determining conditions for thermoplastic processing guaranteeing receipt of high-quality wire rod for cold upsetting using numerical and physical modelling methods, Materials 2020, vol. 13, iss. 3, DOI: 10.3390/ma13030711
  3. Laber K. Innovative Methodology for Physical Modelling of Multi-Pass Wire Rod Rolling with the Use of a Variable Strain Scheme, Materials 2023, vol. 16, iss. 2, DOI: 10.3390/ma16020578.

Strain value and strain rate value were calculated by using numerical modelling- FORGE® Computer Software. This software is based on the Finite Element Method. Below You can find equation that are used for calculation of strain and strain rate value [Transvalor_Solutions/Forge_NxT_2.1/OnlineHelp]:

The equivalent strain rate  is determined using the strain rate tensor. The rheology of a hot

material is heavily dependent on  the expression of which is:

- strain rate tensor

Equivalent plastic strain ?Ì… is determined by integration of the equivalent strain rate as follows:

I don’t want to add these equations to my manuscript because of fact that according to other Reviewer suggestion and comment – the manuscript is too long. So I decided to reduce my manuscript.

Point 5: fig.4 a,b describe the content of the phases, into the picture

Response 5: Description of content of the phases was added into the fig. 4a,b according to Your suggestion.

Point 6: fig.16,17,18 describe the content of the phases, into the picture

Response 6: Description of content of the phases was added into the fig.16,17,18 according to Your suggestion.

Point 7: line 721 "the material after hardening had a martensitic structure" is not correct and what about residual austenite? how many percent?

Response 7: Thank You very much for this comment. You have right. After hardening of steel there are always martensitic structure with some residual austenite. I did not write about it because this part of the research was not the main purpose of the work.  At this moment I was only interested in whether the material deformed at earlier stage of the rolling process had a single-phase structure (austenite) or if was there also ferrite. Because  of fact that in all analysed technological variants, the material after hardening had a martensitic structure with some residual austenite, it can be concluded that, under real conditions (in all the analysed variants), the tested steel was deformed in the RSM rolling block in a single-phase (austenitic) state.

Moreover the content of residual austenite depends on many factors. Among other things, the carbon content in steel for example. Investigated steel (20MnB4) it low carbon steel. To confirm it I made an investigation with using SEIFERT XRD 3003T-T diffractometer. This device allows to detect residual austenite if its content exceeds 4%. Based on the obtained results, it was found that the residual austenite content did not exceed 4% because it was impossible to detect residual austenite in the investigated samples.

I didn’t describe these research in my manuscript because as I wrote earlier – it wasn’t the main purpose of the manuscript. Moreover – according to the Reviewer 2 suggestion – at present  - my manuscript is too long. I must to reduce my paper.

But according to Your comment I modified my manuscript and I added some information about it – according to Your suggestion. Details in the new version of the manuscript.

Point 8: a lot of self-citation.

Response 8: Thank You very much for this comment. According to Your suggestion I modified my paper I removed 7 of my articles and related citations.

Round 2

Reviewer 2 Report

Comments and Suggestions for Authors

Following my previous comments/suggestions for improvement, the author has made several modifications. In fact, the new version is much better. However, before final publishing some details still can be ameliorated:

1) During English language proofreading for final editing, please try to reduce the number of appearances of the string: “on the basis of” (presently it appears 10 times along the manuscript).

2) Table 1 should be mentioned in the text.

3) At lines 148, 266, 346: instead of “Tables 2-3”, write “Tables 2 and 3”.

Comments on the Quality of English Language

During English language proofreading for final editing, some details will be ameliorated.

Author Response

Dear Reviewer

Thank You very much for Your review, comments and suggestion.

Please do not pay attention to the formatting of the article - The article is in the process of formatting.

Below You can find my answers for Your suggestion and comments

Point 1: During English language proofreading for final editing, please try to reduce the number of appearances of the string: “on the basis of” (presently it appears 10 times along the manuscript).

Response 1: I modified my manuscript according Your suggestion. I asked the translation agency to correct the article and limit this phrase. Details in the modified manuscript file.

Point 2: Table 1 should be mentioned in the text.

Response 2: I modified my manuscript according Your suggestion. Details in the modified manuscript file.

Point 3: At lines 148, 266, 346: instead of “Tables 2-3”, write “Tables 2 and 3”.

Response 3: I modified my manuscript according Your suggestion. Details in the modified manuscript file.

Reviewer 3 Report

Comments and Suggestions for Authors

The correction of the article was made partially, please fill in  

line 379 and 705 "with some residual autenite" my question is not how some, but how many % residual autenite, because it will significantly affect the properties

the description of the microstructure is missing, fig. 16 b, 17 b, a higher magnification of the microstructure is needed

Table 2,3,4, write formulas for calculating strain and strain rate , write the formulas, not the literature reference

Author Response

Dear Reviewer

Thank You very much for Your review, comments and suggestion.

Please do not pay attention to the formatting of the article - The article is in the process of formatting.

Below You can find my answers for Your suggestion and comments

Point 1: line 379 and 705 "with some residual autenite" my question is not how some, but how many % residual autenite, because it will significantly affect the properties

Response 1: You have right that residual autenite content has an influence on the properties, but in this case it wasn’t important for me because I was interested in whether the material deformed at earlier stage of the rolling process had a single-phase structure (austenite) or if was there also ferrite. But according to Your suggestion I added an information about residual austenite content to my manuscript.

Point 2: the description of the microstructure is missing, fig. 16 b, 17 b, a higher magnification of the microstructure is needed

Response 2: Figures 15b, 16b and 17b are the figures for the same parameters of rolling process as figures 15a, 16a and 17a. I included photos with a smaller magnification (100 x) to better see the influence of the rolling and cooling process parameters on the banding of the microstructure. I added description of phase to the fig.15b, 16b and 17b according to Your suggestion. Moreover a add an information about magnification value and I paste the figures with larger magnification (200 x)

Point 3: Table 2,3,4, write formulas for calculating strain and strain rate , write the formulas, not the literature reference

Response 3: Dear Reviewer. As I wrote the data presented in these tables were determined in my previous works. To precisely describe the formulas which the Forge® computer software use to determine these parameters, I would have to increase the volume of my manuscript. The most important formulas for calculating strain, strain rate, temperature, etc. should be included. Additionally, literature should be provided. This will increase the volume of the manuscript. However, as another reviewer suggested, the article should be reduced. The data in these tables is provided only for information about rolling conditions. This was not the main purpose of this article. I am asking for acceptance in this form.
